# $Si_{0.97}Ge_{0.03}$ microelectronic thermoelectric generators with high power and voltage densities

Ruchika Dhawan [1,5], Prabuddha Madusanka[1,5], Gangyi Hu[1,3], Jeff Debord[2,4], Toan Tran[2], Kenneth Maggio[2], Hal Edwards[2] & Mark Lee [1✉]

Microelectronic thermoelectric generators are one potential solution to energizing energy autonomous electronics, such as internet-of-things sensors, that must carry their own power source. However, thermoelectric generators with the $mm^2$ footprint area necessary for on-chip integration made from high thermoelectric figure-of-merit materials have been unable to produce the voltage and power levels required to run Si electronics using common temperature differences. We present microelectronic thermoelectric generators using $Si_{0.97}Ge_{0.03}$, made by standard Si processing, with high voltage and power generation densities that are comparable to or better than generators using high figure-of-merit materials. These Si-based thermoelectric generators have $<1\ mm^2$ areas and can energize off-the-shelf sensor integrated circuits using temperature differences $\leq 25\ K$ near room temperature. These generators can be directly integrated with Si circuits and scaled up in area to generate voltages and powers competitive with existing thermoelectric technologies, but in what should be a far more cost-effective manner.

[1] Department of Physics, The University of Texas at Dallas, Richardson, TX 75080, USA. [2] Texas Instruments Incorporated, Dallas, TX 75243, USA. [3]Present address: CGG, Houston, TX 77072, USA. [4]Present address: Microelectronic Devices IP LLC, Dallas, TX, USA. [5]These authors contributed equally: Ruchika Dhawan and Prabuddha Madusanka. ✉email: marklee@utdallas.edu

The development of miniature (<1 cm² total area) silicon integrated circuit (IC) sensors and networking devices for a broad range of internet-of-things (IoT) applications has spurred the question of how to provide reliable and sustainable power to such ICs[1]. IoT devices are often intended to be embedded in enclosed environments not meant to be routinely accessible, such as inside a heating system[2] or buried under pavement[3], where utility line power is unavailable, changing batteries is impractical, and there is insufficient light for photovoltaics. Many IoT devices must then be energy autonomous. That is, they must carry with them a small, renewable energy source, preferably integrated on the same chip or in the same package. Consequently, significant interest has developed in small microelectronic thermoelectric generators (µTEGs) as one method to power energy autonomous IoT devices wherever a reliable thermal gradient exists[1,4–9].

Most current research on thermoelectric (TE) technology concentrates on developing new materials[10] having a high TE figure-of-merit $ZT = (S^2\sigma/\kappa)T$, where $S$, $\sigma$, and $\kappa$ are the material's thermopower, electrical conductivity, and thermal conductivity, and $T = \frac{1}{2}(T_C + T_H)$ is the mean temperature between a cold reservoir at temperature $T_C$ and a hot reservoir at $T_H$ (in Kelvin). This focus on complex high $ZT$ materials is because a TEG's ideal thermodynamic efficiency increases with the $ZT$ of the materials used to form the thermopile[11]. Modern high $ZT$ materials such as PbTe[12], the BiSbTe system[13,14], CuI[15], Heusler alloys[16], $SnS_{1-x}Se_x$[17], $CsSnI_{3-x}Cl_x$[18], $Cu_2Te:Ga$[19], and dichalcogenides[20] generally aim to achieve $ZT \approx 1$ for $T$ near 300 K.

Higher efficiency means less heat is drawn to generate a given power. Maximizing efficiency is important if the total heat capacities of the $T_H$ and $T_C$ reservoirs are small enough that the heat flow from $T_H$ to $T_C$ significantly decreases the temperature difference $\Delta T = (T_H - T_C)$. However, for µTEGs the heat flow cross-section is small, so little heat is typically drawn, and the $T_H$ and $T_C$ heat capacities are usually very large or have actively maintained temperatures. In this case efficiency may not be the primary concern. The critical criterion is the ability to directly energize an IoT device or trickle charge its battery when operating from commonly encountered $\Delta T$s between 10 to 50 K with $T_C$ near room temperature. In practice this means generating voltage >1.5 V with ≥ several µA of current (i.e., several µW of power). This voltage is required to cross the threshold that turns on Si transistors or to push charge into a typical battery. Because material Seebeck coefficients are typically ~0.1 mVK$^{-1}$, producing >1.5 V from $\Delta T = 10$ K requires a thermopile connecting ~$10^3$ thermocouples in series. TEGs using bulk high $ZT$ materials need areas of several cm² to accommodate this many thermocouples[21]. Small area (≤ few mm²) high $ZT$ TEGs, which are desirable for integration with IoT devices, have yet to reach this voltage/current threshold using $T_C$ near 300 K and moderate $\Delta T$ ~ 20 K[4,6,7]. Furthermore, high $ZT$ materials can be expensive to synthesize, often contain toxic or non-earth-abundant elements[15,17], and are incompatible with Si IC processing, all of which increase the cost-per-Volt and cost-per-Watt generated.

In this article we report small area (≪1 mm²) µTEGs with $Si_{0.97}Ge_{0.03}$ as the TE material, fabricated using standard Si IC processing. These µTEGs can generate power densities (per unit area for heat flow) comparable to or better than high $ZT$ TEGs and can energize IoT devices from commonly encountered $\Delta T$s. These µTEGs build on the alternative approach to Si-based µTEGs we recently reported[22] to overcome silicon's inferior $ZT$[23]. This approach emphasizes application of device physics and circuit engineering principles to optimize a µTEG's generated power density at given $\Delta T$, rather than focusing on thermodynamic efficiency. This strategy exploits the ability of Si processing to fabricate thermopiles consisting of a very large number of TE elements in a small area, thereby producing a high total power density despite relatively low power per TE element, and to control parasitic thermal and electrical resistances.

## Results

**Description of µTEG device structures**. Two types of µTEG devices were made, test mode and harvest mode, all fabricated on an industrial 65 nm node Si IC process line. The test mode device structures and measurement protocols are identical to those detailed in refs. [22,24]. Design and fabrication details for the harvest mode devices are given in Methods and in Supplementary Fig. 1. Each test mode device constitutes a thermocouple having total cross-sectional area of 48 µm × 36 µm with an on-chip integrated resistive heater as the $T_H$ reservoir. The purpose of the integrated heater is to give a highly reproducible series thermal impedance between heat source and thermocouple. This facilitates de-embedding the thermocouple's intrinsic performance characteristics from parasitic thermal impedances. However, most µTEG applications require harvesting heat from an off-chip $T_H$ source. Harvest mode µTEGs omit the integrated heater and instead connect a thermopile thermally (but not electrically) to a thermal contact pad on the chip surface. A heated copper rod placed on this pad acts as the $T_H$ reservoir, so the thermal impedance depends sensitively on the quality of the contact between Cu rod and thermal pad.

Operating from $T_C$ near 300 K and $\Delta T$ between 5 to 50 K, test mode µTEGs were designed to optimize power density, not voltage. By contrast, harvest mode µTEGs were designed to maximize voltage density rather than power and so consist of 640 thermocouple unit cells (each with area of 19.8 µm × 15.7 µm) connected in series. As the following results show, operating from nearly the same $T_C$ and $T_H$, test mode devices generated power density ~6× higher than harvest mode, while harvest mode devices generated voltage density ~3.6× higher than test mode.

The basic TE elements of both test mode and harvest mode devices are 80 nm wide × 700 nm long × 350 nm tall blades of $Si_{1-x}Ge_x$, where $x$ is nominally 0, 0.01, 0.02, and 0.03. To maintain compatibility with standard Si IC processing, bulk $Si_{1-x}Ge_x$ could not be used. Instead, as described in Methods, Ge was incorporated into the top surface of a 300 mm diameter Si wafer by ion implantation followed by activation anneal. For reasons given in Methods, this restricted the maximum usable Ge concentration to $x \leq 0.03$.

$Si_{1-x}Ge_x$ was used because both bulk and nanostructured $Si_{1-x}Ge_x$ show significantly enhanced $Z$ compared to pure Si due to suppression of the phonon contribution to $\kappa$ through random alloy and grain boundary scattering[25]. A large amount of TE device work using $Si_{1-x}Ge_x$ exists, particularly targeted at high temperature applications[25–29]. These works generally use alloy compositions with $0.2 \leq x \leq 0.5$ because $\kappa$ is near its minimum value through that range[25,30,31]. However, the majority of the decrease in $\kappa$ with increasing $x$ occurs in the narrow range going from $x = 0$ to $x \approx 0.05$[25,30–33]. This suggests that a significant increase in $Z$ and hence TE performance may be expected using only a few % Ge.

**Performance characteristics of test mode µTEGs**. Figure 1a shows power–current–voltage (P–I–V) characteristics at various $\Delta T$ ($T_C = 300$ K) of the test mode $Si_{0.97}Ge_{0.03}$ µTEG with the highest power density. The thermopile design for this specific µTEG is given in Supplementary Fig. 2. Three nominally identical devices were tested; all had P–I–V characteristics within 5% of each other. As $\Delta T$ increases, the linear I–V offsets further from the origin. The source resistance is $R_S = |\Delta V/\Delta I| = 5.2$ Ω. The open-circuit voltage, $V_{OC}$, and short-circuit current, $I_{SC}$, are the intercepts of the I–V lines with

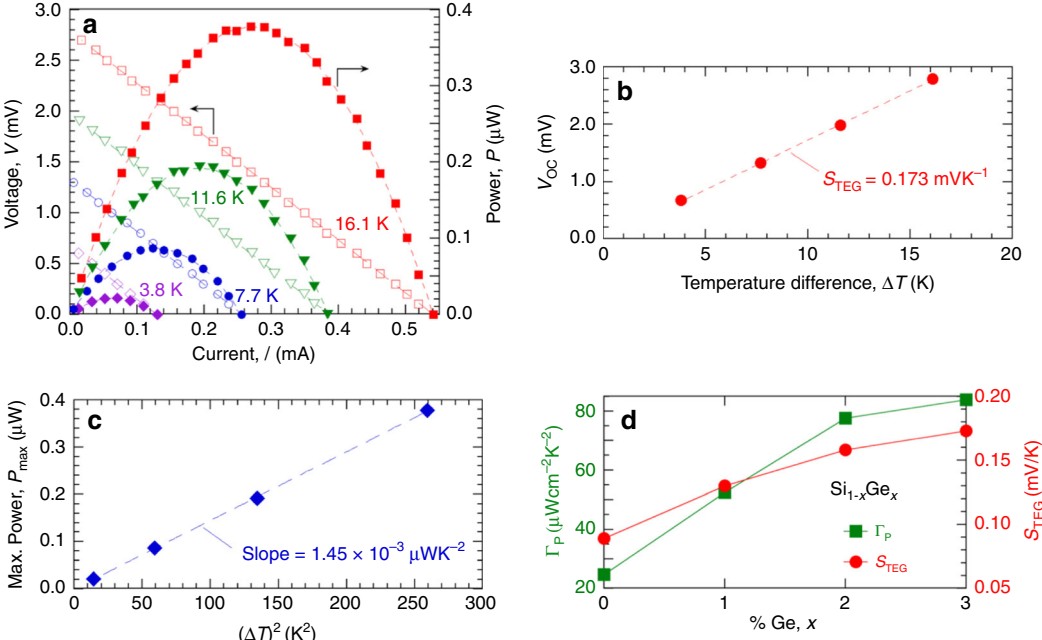

**Fig. 1 Performance of test mode µTEG with highest power density. a** Power–current–voltage data with $T_C = 300$ K and $\Delta T = 3.8$ K (purple diamonds), 7.7 K (blue circles), 11.6 K (green triangles), and 16.1 K (red squares). Open symbols are voltage data (left axis) and filled symbols are power $= V \cdot I$ data (right axis). Dashed lines are linear (for $I–V$) and quadratic (for $I–P$) least-square fits to the data. **b** Open circuit voltage $V_{OC}$ vs. $\Delta T$. The dashed line is a linear least-square fit. **c** Maximum power, $P_{max}$, as determined from the data in **a**, vs. $(\Delta T)^2$. The dashed line is a linear least-square fit. **d** Specific power density, $\Gamma_P$ (green squares, left axis) and TEG device Seebeck coefficient, $S_{TEG}$ (red circles, right axis) vs. Ge percentage $x$ for four µTEGs having the same device layout as the one represented in **a**. The solid lines simply connect data points.

the $V$ and $I$ axes, respectively. The generated power $P = VI$ has maximum $P_{max} = V_{OC}I_{SC}/4 = V^2_{OC}/4R_S =$ power delivered to a load resistance $R_L = R_S$, known as matched load conditions. Figure 1b shows $V_{OC}$ is linearly dependent on $\Delta T$, with the slope of the linear fit giving the Seebeck coefficient of the TEG device, $S_{TEG} = V_{OC}/\Delta T = 0.173$ mVK$^{-1}$. Figure 1c shows $P_{max}$ is linearly dependent on $(\Delta T)^2$. The slope of the fitted line $= 1.45 \times 10^{-3}$ µWK$^{-2}$ gives the power per square of temperature difference. Normalizing to the 48 µm × 36 µm heat flow cross-sectional area gives the specific power density, $\Gamma_P = 84$ µWcm$^{-2}$K$^{-2}$. $\Gamma_P$ measures $P_{max}$ normalized to both TEG area and operating $\Delta T$. Figure 1d plots how $\Gamma_P$ and $S_{TEG}$ increase with $x$ for four µTEGs having the same design as the µTEG of Fig. 1a, but different $x$. For this µTEG design, $\Gamma_P$ increases by a factor of 3.5× and $S_{TEG}$ approximately doubles as $x$ goes from 0 to 0.03. $\Gamma_P$ does not exactly scale with $S^2_{TEG}$ because $R_S$ increases by ~10% with Ge content over this range.

For the TEG device $V_{OC} = S_{TEG}\Delta T$, but at the level of the thermopile itself, $V_{OC} = S\Delta T_{TP}$, where $S$ is the net Seebeck coefficient of the TE material and $\Delta T_{TP}$ is the actual temperature difference across the TE blades forming the thermopile. Because of parasitic thermal impedances between hot/cold reservoirs and the TE blades, $\Delta T_{TP} < \Delta T$, and for pure Si ($x = 0$) thermopiles we estimated[22] that $\Delta T_{TP}/\Delta T \approx 0.10$ to $0.18$. For Si$_{1-x}$Ge$_x$, literature values show that the TE material $S$ is insensitive to $x$ between $x = 0$ and $0.03$[25,31]. Consequently, the increase in $S_{TEG}$ with $x$ from Fig. 1d indicates that $\Delta T_{TP}$ must nearly double (at same applied $\Delta T$) as $x$ increases from 0 to 0.03 due to a decrease in TE material $\kappa$ with increasing Ge content.

For each value of $x$, we tested sixteen µTEG layout design variations. Layout structure variations explored different number of TE blade elements per unit area, different electrical lead and contact configurations, and different heat exchange structures to thermally couple to the $T_H$ reservoir, but all used the same TE blade size and n- and p-dopant densities. For any given layout,

$\Gamma_P$ increased monotonically with increasing $x$, with $\Gamma_P(x = 0.03)/\Gamma_P(x = 0) = 2.5$ to $3.5$ depending on layout design. Among the 16 different µTEG layouts with $x = 0.03$, the variant used for Fig. 1a gave the highest $\Gamma_P$, the variant with the lowest $\Gamma_P$ generated 5 µWcm$^{-2}$K$^{-2}$, and the plurality of layout variants gave $\Gamma_P$ between 20 to 30 µWcm$^{-2}$K$^{-2}$. Higher $\Gamma_P$ layouts were associated with two features. First, they had electrical and thermal lead/contact configurations that gave lower parasitic series resistances. Second, they came closer to using an optimum number of TE blade elements to maximize $V^2_{OC}/R_S$ by properly balancing the trade-off between using fewer TE blades to increase the thermopile's thermal resistance to increase $\Delta T_{TP}$ and hence $V_{OC}$, and using more TE elements to decrease the thermopile's $R_S$[24,34].

In situations where the thermal reservoirs have large heat capacities or where $T_H$ and $T_C$ are actively maintained, $\Gamma_P$ may be a more practically important metric than efficiency. $\Gamma_P$ can be used to compare power generation capability across different types of TEGs. For example, from its data sheet[21] a high $ZT$ TEG of 9 cm$^2$ area generates $P_{max} = 0.41$ W from $T_H = 110\,°C$ and $T_C = 50\,°C$, so its $\Gamma_P = 12.7$ µWcm$^{-2}$K$^{-2}$. $\Gamma_P$ values compiled from summaries[7,35–37] of (Bi,Sb)$_2$(Te,Se)$_3$ TEGs range from 1 to 20 µWcm$^{-2}$K$^{-2}$ for commercial devices and up to ~100 µWcm$^{-2}$K$^{-2}$ for research devices. Thus, the $\Gamma_P = 84$ µWcm$^{-2}$K$^{-2}$ for our Si$_{0.97}$Ge$_{0.03}$ µTEG is competitive with the best high $ZT$ TEGs from the standpoint of areal power density produced using the same $\Delta T$.

**Performance characteristics of harvest mode µTEGs.** Figure 2a illustrates the cross section of a harvest mode µTEG. Details of the harvesting µTEG measurement protocol are given in Methods. The top of a harvest mode thermopile is thermally connected (but electrically isolated) through an integrated heat exchanger to an Al coated thermal contact pad, shown in Fig. 2b. The heat exchanger consists of several layers of interdigitated Cu

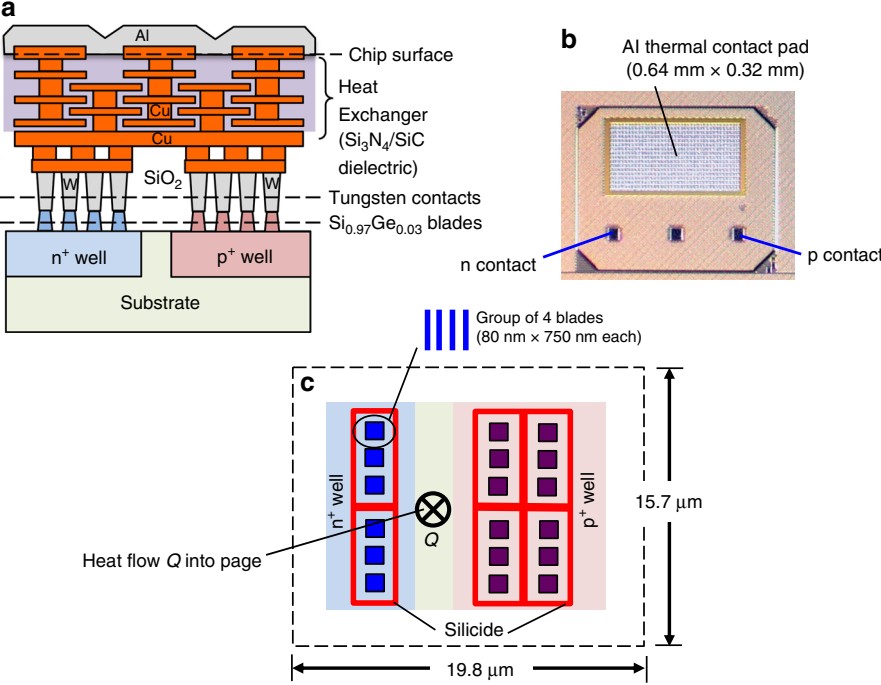

**Fig. 2 Design of a harvest mode μTEG. a** Illustration (not to scale) of the side-view cross section through one n-p thermocouple with contact metallization and heat exchanger layers to a surface aluminum thermal contact pad. **b** Optical microscope plan view image of the Al coated thermal contact pad, with electrical contacts to the n and p sides of the thermopile array. (Middle contact pad is to the substrate and is not used.) **c** Plan view design (to scale) looking down on one thermopile unit cell forming the harvest mode μTEG used to generate the data of Fig. 3. Each dark-colored solid square represents a group of four blade elements. The dark red lines are silicide electrical contacts to the n$^+$ and the p$^+$ wells. As depicted, the thermal contact would be above the page and the substrate behind the page, so heat $Q$ flows perpendicularly into the plane of the page as indicated while electric current flows from n$^+$ well to p$^+$ well through the metal contacts bridging the n- and p-sides shown in part (**a**).

electrodes, one set extending up from the thermopile and the other extending down from the thermal contact pad, spaced by a dielectric stack consisting of relatively high thermal conductivity Si$_3$N$_4$/SiC layers.

Harvest μTEGs were designed to generate high voltage density rather than high $\Gamma_P$, so they consist of many small thermocouple unit cells connected electrically in series and thermally in parallel. Figure 2c depicts the design of one such unit cell. Each unit cell is built using the same size, shape, and dopant density TE blade elements as test mode devices, but has fewer blades per unit area to facilitate the multiple series electrical connections needed to increase output voltage. Since the n-side blades are connected electrically in parallel, as are (separately) the p-side blades, fewer blades result in higher resistance per unit area and hence lower output current and power density. A complete harvest mode μTEG is composed of 640 unit cells covering a total heat flow cross-sectional area of 0.64 mm × 0.32 mm, the same as the surface Al thermal contact pad.

Figure 3a shows $P–I–V$ characteristics of a Si$_{0.97}$Ge$_{0.03}$ harvest mode μTEG whose unit cell design is depicted in Fig. 2c. A heated Cu rod touching the thermal contact pad served as the $T_H$ source. Details of the measurement protocol are given in Methods. Figure 3b plots $V_{OC}$ vs. $\Delta T$ to obtain the total Seebeck coefficient $S_{tot} = 0.102$ VK$^{-1}$ for the 640 unit cells in series. We found $S_{tot}$ could vary between 0.07 to 0.11 VK$^{-1}$ depending strongly on how well the Cu rod contacted the thermal pad. From Fig. 3b, the Seebeck coefficient per cell is then $S_{cell} = S_{tot}/640 = 0.16$ mVK$^{-1}$. The source resistance of this harvesting μTEG is $R_S = 76$ kΩ. Among harvesters tested of identical design, $R_S$ was between 75 to 77 kΩ independent of Cu rod contact conditions. The resistance per unit cell is $R_{cell} = R_S/640 = 120$ Ω. The harvester's $R_{cell}$ is greater than the test mode's $R_S$ because the test mode

thermocouple consists of 20× more TE blades connected in parallel, reducing the test mode's source resistance and increasing its $I_{SC}$ compared to the harvest device. If we scale $R_{cell}$ to the same number of blades in parallel as the test mode device, the harvester's per-cell resistance would then be $R_{cell}/20 = 6$ Ω, slightly more than the $R_S = 5.2$ Ω for the test mode device from Fig. 1a. Previous modeling[22,24] of $x = 0$ test mode devices estimated the parasitic resistance from leads and contacts to be ~2 Ω per thermopile. Harvest mode devices may have a somewhat higher parasitic resistance per cell due to the additional leads and contacts needed to connect multiple thermocouple cells in series, connections not needed in test mode device.

**Energizing IoT devices.** Using $\Delta T$ from 20 to 25 K, these harvest μTEGs could energize commercial Si ICs made for low-power IoT applications. Figure 4a illustrates a harvest μTEG connected as the unregulated power input to a BQ25570 power management integrated circuit (PMIC)[38]. PMICs are widely used to support energy autonomous electronics by producing a regulated output voltage $V_{out}$ from an unregulated, high source resistance input $V_{in}$ and storing excess input energy by charging a capacitor or back-up battery. The BQ25570 was run without a battery and so used only the electrical input from the μTEG. To initiate a cold start from the state where the PMIC is fully discharged required operating the μTEG with $\Delta T = 29$ K to charge the PMIC's storage capacitor up to $V_{stor} = 4.2$ V. This stored charge is used to regulate $V_{out}$. After cold start, the PMIC operated continuously with $\Delta T$ as low as 24 K. Figure 4b plots the PMIC's steady-state $V_{out}$ and $V_{stor}$ vs. load resistance $R_L$, with μTEG operating from $\Delta T = 24$ K. The PMIC was configured to produce a regulated $V_{out} = 1.80$ V, which it could do for $R_L \geq 0.900$ MΩ, corresponding to a maximum output current of 2 μA. For $R_L < 0.900$ MΩ the load's

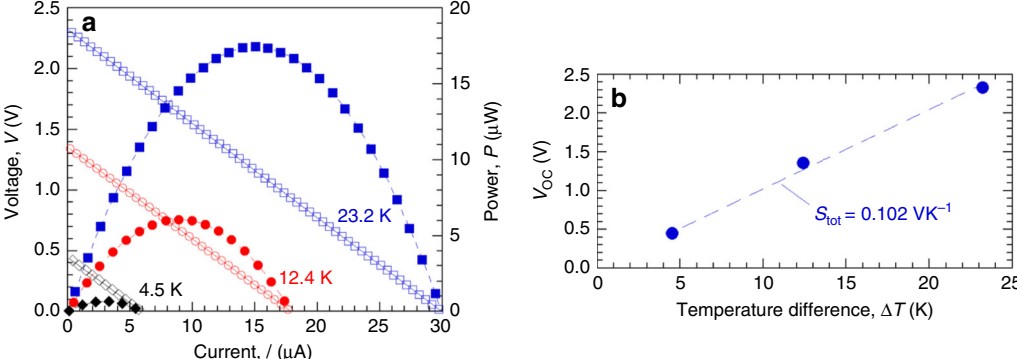

**Fig. 3 Performance of a harvest mode µTEG. a** Power–current–voltage data with $T_C = 300$ K and $\Delta T = 4.5$ K (black diamonds), 12.4 K (red circles), 23.2 K (blue squares). Open symbols are voltage data (left axis) and filled symbols are generated power $= V \cdot I$ data (right axis). Dashed lines are linear (for $I$–$V$) and quadratic (for $I$–$P$) least-square fits to the data. **b** Open circuit voltage $V_{OC}$ vs. $\Delta T$. The dashed line is a linear least-square fit.

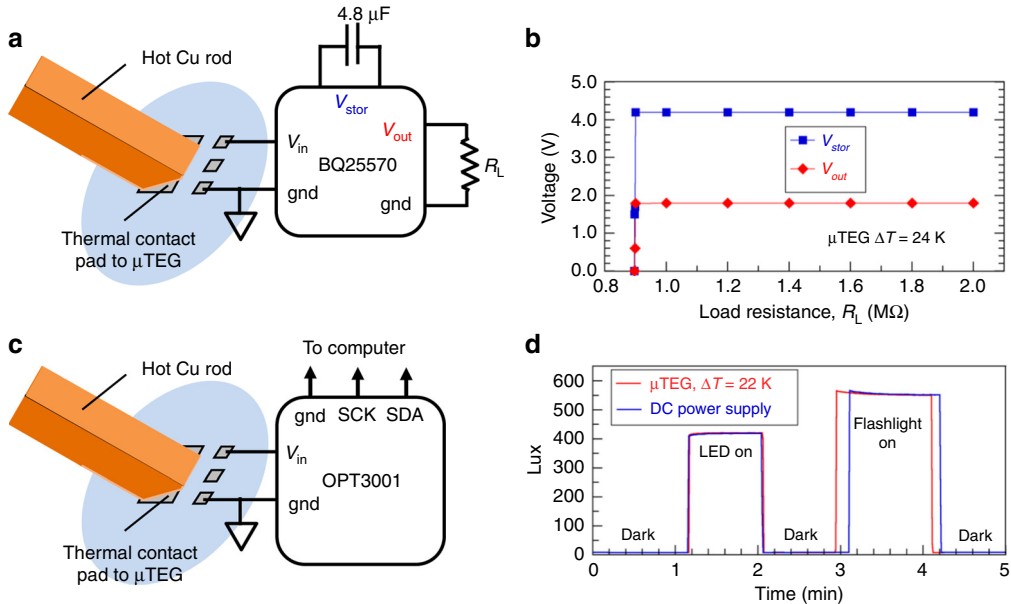

**Fig. 4 Performance of integrated circuits energized by a harvest mode µTEG. a** Illustration of the connections between a hot Cu rod, thermal contact pad, µTEG electrical contacts, and a BQ25570 power management integrated circuit (PMIC) with a 4.8 µF storage capacitor and a load resistance $R_L$ at its output. **b** PMIC steady-state storage voltage (blue squares) and output voltage (red diamonds) vs. $R_L$, µTEG operating from $\Delta T = 24$ K. **c** Illustration of the connections between a hot Cu rod, thermal contact pad, µTEG electrical contacts, and an OPT3001 ambient light sensor. **d** Ambient light intensity sensed by the OPT3001 in response to a red light-emitting diode (LED) and a white flashlight. The red line is the signal when energized directly by the µTEG operating from $\Delta T = 22$ K. The blue line is the signal when energized by a standard DC power supply.

current demand outpaced the ability of the µTEG to supply power, forcing the PMIC to discharge $V_{stor}$ thus driving $V_{out}$ to zero. If this µTEG/PMIC configuration were energizing a real device having a variable load resistance, the device would either be fully on (when $R_L > 0.90$ MΩ) or fully off (when $R_L < 0.90$ MΩ), as Fig. 4b shows a very sharp transition between $V_{out} = 1.80$ V and $V_{out} = 0$ V. In a real situation, operational continuity would be maintained when $R_L$ drops below 0.90 MΩ (or when $\Delta T$ drops to <24 K) by using a backup battery with the PMIC.

Figure 4c illustrates a harvest µTEG connected directly to energize a commercial OPT3001 visible light sensor intended for use as an IoT sensor[39]. The sensor's data sheet specifies a minimum input voltage and current of 1.6 V and 1.8 µA. We first powered the OPT3001 using the µTEG via the PMIC output, and it ran stably using $\Delta T = 24$ K. Because the PMIC needs to draw power to perform its regulation functions, we tried powering the OPT3001 directly from the µTEG and found it operated within specified tolerances using $\Delta T$ down to 22 K. This was the smallest

$\Delta T$ at which the harvester could generate both the minimum voltage and the minimum current needed to for the OPT3001 to operate within specifications. (From Fig. 3b, using $\Delta T = 17$-18 K would generate $V_{OC} = 1.8$ V, the same as the PMIC output voltage used to energize the OPT3001, but at zero current.) Fig. 4d shows the OPT3001's light intensity readings when powered by the µTEG is identical to its readings when powered by a conventional DC power supply. Further details on the operation of the BQ25570 and OPT3001 using the µTEG are given in Methods.

## Discussion
The µTEGs in this work use small footprint areas appropriate for on-chip or in-package integration with energy autonomous IoT ICs. Given the highly parallel nature of Si fabrication over a 300 mm diameter Si wafer, there are no significant technical barriers to scaling such µTEG designs to much larger areas. For example, using an appropriate photolithography mask set, Si IC fabrication

could make over $5 \times 10^5$ replicas of the thermopile used for Fig. 1 within a 9 cm$^2$ area without adding processing steps or increasing process time or cost. Comparing to the same area bulk high $ZT$ material TEG of ref. [21] operating from the same $\Delta T = 60$ K, a Si$_{0.97}$Ge$_{0.03}$ TEG would generate optimal power of 2.7 W compared to the 0.41 W for the bulk TEG. Perhaps as importantly, industrial Si processing uses widely abundant materials and has a much higher production volume throughput than any other material technology, so the cost-per-watt generated with a Si-based TEG should be substantially lower than with any other TE material.

All our μTEG devices were designed to be tested on a wafer probe station with the probe station chuck as $T_C$ reservoir, so both thermal interfaces were incompatible with standard IC chip package heat exchangers. Looking towards the future, engineering thermal interfaces to optimize heat exchange between a μTEG's hot and cold thermal contacts and application-specific $T_H$ and $T_C$ reservoirs will be critical to advancing practical use of μTEGs in energy autonomous devices. The goal is to minimize parasitic series and contact thermal impedances and to maintain uniform heat flow through the μTEG thermopile cross section. Low thermal impedance chip packages[40] designed to remove heat from power ICs to a cold reservoir could conceivably be adapted for use with a μTEG's cold side contact. Solutions for the hot side contact are less straightforward as there is little established work aimed at directing external heat into an IC chip.

Assuming thermal interface issues can be solved, these Si based μTEGs could energize IoT ICs and sensors using a $T_C$ near 300 K and $\Delta T$ of 20 to 25 K. Several conceivable IoT environments can generate such temperature profiles, such as the temperature differences between the exterior ($T_C \sim 273$ K) and interior ($T_H \sim 295$ K) of a heated building in winter, or between subsoil earth ($T_C \sim 285$ K) and roadway pavement ($T_H \sim 310$ K)[3]. Using μTEGs for biothermal energy harvesting presents a more difficult challenge, since $\Delta T$ between core human body temperature and an air-conditioned room is about 10 to 15 K, and $\Delta T$ between skin surface temperature and ambient air is usually taken to be ≤ 5 K[41]. Because TEG power generation scales as $(\Delta T)^2$, reducing $\Delta T$ from 20 K to 5 K using the same TEG device reduces power output by a factor of 16. The Si based harvest mode μTEGs presented here could compensate for that power reduction by increasing area by a factor of 16. Using the same harvesting μTEG design of Fig. 2 would then require a total μTEG area of $16 \times 0.2$ mm$^2$ = 3.2 mm$^2$, not too much larger than the 1 mm$^2$ desired for integrated energy autonomous devices. This area could be further reduced by increasing the number of TE blade elements in each unit cell of this harvesting mode μTEG design.

## Methods

**General μTEG design and processing.** All μTEGs were fabricated on an industrial 65 nm node technology silicon complementary metal-oxide-semiconductor (CMOS) process line on a 300 mm diameter Si (100) oriented wafer. Designs complied with all standard design rules, including minimum feature areas, line-widths, and aspect ratios, and used only material sets and dopants normally available for commercial Si CMOS device fabrication. These design rules ensure process compatibility with all other CMOS devices and circuits that could be fabricated on the same wafer.

The front surface of each blank wafer was protected with a 50 nm thick thermal oxide. Then a thin surface Si$_{0.97}$Ge$_{0.03}$ alloy layer was created using a blanket (unmasked) Ge ion implantation followed by activation anneal. Three consecutive implant energies & dosages were used to form a Si$_{0.97}$Ge$_{0.03}$ layer: (1) 100 keV & $1.2 \times 10^{16}$ cm$^{-2}$, (2) 200 keV & $6.0 \times 10^{15}$ cm$^{-2}$, and (3) 270 keV & $2.4 \times 10^{16}$ cm$^{-2}$, followed by a 1050 °C activation anneal for 20 mins. Simulations of Ge density vs. depth into the wafer surface are shown in Supplementary Fig. 3. The freely available Monte-Carlo based Transport of Ions in Matter (TRIM) application[42] was used to model the as-implanted Ge distribution, but it does not simulate annealing. A Technology Computer Aided Design (TCAD) semiconductor process simulator[43] was also used to estimate implanted Ge distribution after annealing, using published values of thermal diffusion coefficients for Ge in Si[44]. Results

indicate the Ge density is between 1 to $2 \times 10^{21}$ cm$^{-3}$ to a depth of $\sim 250$ nm. Nominal 3% Ge corresponds to a Ge density of $1.5 \times 10^{21}$ cm$^{-3}$, and the base of the "blades" that form the thermopile structure are etched down to a nominal depth of 350 nm.

Post-anneal optical microscope inspection using a Schimmel defect etch and stain[45] showed no detectable defects resulting from the implantation. However, for $x > 0.03$ the surface Si$_{1-x}$Ge$_x$ layer resulted in sufficient bowing of the wafers that the wide area, very high resolution, shallow depth-of-focus photolithography needed could no longer be done with adequate precision. This prevented us from going higher than 3% Ge content.

The fundamental thermopile elements were nanostructured blades formed by the same photolithographic masking and Si etch process normally used to create isolation trenches for Si CMOS transistor circuits in this process technology. Doped n-type blades were etched from n$^+$-wells formed by P and As ion implantation (dopant concentration $3.9 \times 10^{18}$ cm$^{-3}$), and p-type blades were etched from p$^+$-wells formed by B ion implantation (dopant concentration $4.3 \times 10^{18}$ cm$^{-3}$). Each individual blade was nominally 80 nm wide × 750 nm long × 350 nm tall, although cross-sectional scanning electron microscope (SEM) images[22] showed the actual blades to be slightly trapezoidal in cross section. An 80 nm width was used as it is the minimum width that can be reliably etched to form a 3-dimensional structure using 65 nm node process technology. SiO$_2$ filled the space between blades for mechanical support. Each blade was electrically and thermally contacted individually from the top using a tungsten (W) plug. The blades were electrically contacted from the bottom using communal n$^+$- and p$^+$-well contacts formed by a mesh of silicide lines in each well. The silicide mesh was used to minimize the parasitic series spreading resistance through the relatively high resistivity doped silicon wells to the metal electrodes.

In all test mode μTEGs and in each unit cell of a harvest mode μTEG, Cu metal layers and vias were used to connect all n-type blades electrically in parallel, and, separately, all p-type blades electrically in parallel. The n-type side and the p-type side were then connected electrically in series to form a thermopile.

**Test mode μTEG design.** A detailed plan view design illustration of the particular thermopile layout of the test mode μTEG used to generate the data in Fig. 1 of this paper can be found in Supplementary Fig. 2.

**Harvest mode μTEG design.** A design illustration of one thermopile unit cell of the harvest mode μTEG used to generate the data in Fig. 3 of this paper, including the Cu metal layers used to electrically connect the TE blade elements, can be found in Supplementary Fig. 1. Each thermopile unit cell is assigned a border area of 19.8 μm × 15.7 μm. The complete harvest mode μTEG used for Fig. 3 of the paper consists of 640 such thermocouple unit cells, electrically connected in series, arranged in a 40 cell × 16 cell array, occupying an area of 0.32 mm × 0.64 mm. The surface Al coated thermal contact layer is formed directly over the footprint of this array.

**Harvest mode μTEG measurement procedure.** The original 30 cm diameter processed wafer was diced into 2 cm × 3 cm die, each die containing many test mode and harvest mode μTEG devices. A die was placed on a gold-plated copper chuck in an enclosed electrical probe station. A thin layer of thermal grease applied to the underside of the die was used to improve thermal contact to the chuck. A calibrated platinum resistor thermometer embedded in the chuck monitored chuck temperature (used as $T_C$ in μTEG measurements), and another calibrated thermometer in the probe station monitored ambient environmental temperature. Both temperatures were recorded using a Lakeshore 336 temperature controller. Electrical contact to the n- and p-contact pads shown in Fig. 2b was made using 10 μm radius beryllium copper probe tips to form a 2-probe contact configuration to measure the μTEG current–voltage (I–V) characteristics. All I–Vs were measured with an Agilent 4156 C semiconductor parameter analyzer set to voltage bias from −2 to +2 V. The I-V of the μTEG was always first measured with no heat source applied to the thermal contact pad to establish equilibrium ($\Delta T = 0$) electrical characteristics.

A heated rod made of oxygen-free high conductivity (OFHC) copper brought into physical contact with the Al thermal contact pad was used as the hot reservoir ($T_H$). The Cu rod was ohmically heated using nickel chromium (NiCr) wire (insulated with polyimide) wrapped tightly around the rod. The diameter of the Cu rod was tapered in stages down to a polished flat that approximated the area of the thermal contact pad. The rod was mounted in a probe station micro-manipulator to land on the thermal contact pad. Buffering the contact pad with a small amount of pure indium, first mechanically pressed onto the thermal contact pad and then flowed briefly using a low-temperature soldering iron, was found to enhance thermal contact between the Cu rod's flat and the thermal contact pad.

After touching down the Cu rod onto the thermal contact pad and electrically biasing the rod's NiCr heating element, the temperature $T_H$ was measured by touching the tip of a standard type-K digital thermometer (with NIST-traceable calibration) to the Cu rod as close to the thermal contact pad as mechanically feasible. This same digital thermometer was also used to check the temperature of the probe station chuck where the Si die met the chuck surface. This measurement

of chuck temperature always agreed with the chuck's embedded thermometer to within ±0.2 K, so the chuck's embedded thermometer was used to determine $T_C$.

**Integrated circuit measurement protocol.** Both the BQ25570 and the OPT3001 ICs were purchased solder-mounted onto evaluation module (EVM) printed circuit boards. The EVMs brought the ICs' input and output pins out to convenient wiring terminals and provided resistor networks and jumpers to select various function settings. For both ICs, the power input ($V_{in}$) terminal on the EVM was wired directly to the probe station probe contacting the n-contact on a harvesting mode μTEG, and the circuit common (GND) terminal on the EVM was wired directly to the probe station probe contacting the p-contact on the same harvesting mode μTEG. Total external wiring resistance was <2 Ω, negligible compared to the 76 kΩ resistance of the μTEG.

For the BQ25570, no back-up battery was used. Energy was stored using the 4.8 μF storage capacitor that came mounted on the EVM. Without a back-up battery and with no charge on the storage capacitor, the BQ25570 needed to be "cold-started" by first charging the storage capacitor before it began delivering output power. The cold start needed a minimum $\Delta T = 29$ K applied to the μTEG. Settings on the BQ25570 were configured so that it began delivering regulated output voltage of 1.80 V when the voltage on this storage capacitor reached 4.2 V (its minimum setting), which ended the cold start phase. After cold start, the BQ25570 delivered a steady-state regulated 1.80 V output with a $\Delta T = 24$ K applied to the μTEG. The BQ25570 was also configured to maximize power input from the μTEG by dynamically adjusting its input resistance to match the μTEG's source resistance, thereby transferring $P_{max}$ from the μTEG. Finally, the output terminals on the EVM were directly wired to a variable MΩ resistor box used to vary the load resistance. Voltmeters monitored the output voltage across the resistor box as well as the voltage across the storage capacitor to generate the data in Fig. 4b of this paper.

For the OPT3001, the EVM normally connects to a computer through a USB interface that powers the sensor and sends serial digital data from the sensor to the computer. The data is processed by an executable program[46] into light intensity (in units of lux). For this experiment, the $V_{in}$ terminals on the EVM were not connected to the USB interface but were instead wired to the μTEG either via the BQ25570 or directly to the μTEG. The serial data links remained connected to the USB interface. The OPT3001 EVM was mounted in fixed position inside an opaque box with a red light emitting diode (LED) light source inside the box and a portal through which a white flashlight could be shone onto the sensor. Light intensity levels were recorded in the dark, with the red LED on, and with the flashlight on.

The OPT3001 operated stably through the BQ25570 using $\Delta T$ as low as 24 K applied to the μTEG, or directly from the μTEG using $\Delta T$ as low as 22 K. To compare whether the measured light intensities were reliable when using the experimental thermoelectric energy source, we repeated light measurements with the sensor energized using a conventional wall-plug powered DC power supply using the voltages specified in the sensor's technical data sheet.

## Data availability

The data that support the findings of this study are available from the corresponding author on reasonable request.

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

## Acknowledgements

We thank Ricardo Rivera-Matos for instruction on low-power management and voltage regulation techniques and advice on how to use the BQ25570 with a μTEG input, David Wyke for performing the defect stain inspection of a post-annealed Ge implanted Si wafer to verify absence of crystal damage, and Drew Edwards for running TRIM simulations. This work was supported by the United States National Science Foundation (Grant No. ECCS-170758) and by Texas Instruments Incorporated.

## Author contributions

R.D., P.M., and G.H. carried out the experiments, analyzed results, and helped write the paper. H.E. conceived the nanoblade μTEG devices. K.M. and H.E. designed the device layouts. K.M. performed thermal simulations on the designs. H.E., J.D., and T.T. developed processing recipes and supervised fabrication of the wafers. H.E. and M.L. analyzed results and conceived further experiments and measurements. M.L. designed and set up measurements, performed calculations, and drafted the paper. All authors coedited the manuscript.

## Competing interests

T.T., K.M., and H.E. are employed by Texas Instruments Incorporated. J.D. performed this work as an employee of Texas Instruments Incorporated. The remaining authors declare no competing interests.
