## [Peer Review File · Nature Communications]

Editorial Note: Parts of this peer review file have been redacted as indicated to remove third-party material where no permission to publish could be obtained.

REVIEWER COMMENTS

Reviewer #1 (Remarks to the Author):

This is a sequel to the authors' previous work of Ref.23 and 25. The novelties of this work are the inclusion of Ge into Si thermoelements and the demonstration operation of their micro-TEG device by an external heat source to drive off-the-shelf power management IC and light sensor. I can support the authors' approach to optimize the power density at given temperature difference, rather than focusing on thermodynamic efficiency. This paper will attract interest of scientists in the areas of thermoelectric energy harvesting.

It is a notable result that the specific power density was improved significantly by doping small amount of Ge up to 3 % in Si blade. However, it is to be regretted that there is no discussion on the scientific reasoning. I think there is no reason not to show the dependence of Seebeck coefficient on the Ge concentration. I expect that the authors provided test element group devices to measure the resistivity and carrier concentration of SiGe blades. These data are essential to deepen understanding of the effect of Ge doping.

I am interested in the the statement in line 52, "It has proven difficult to microfabricate high ZT materials..." The authors might want to mention the reason for this statement, or show supporting references.

Reviewer #2 (Remarks to the Author):

The design, fabrication, and analysis of performance keypoints is innovative, under my point of view. All these things are meaningful and very different from the classical TE-devices.

The manuscript is a follow-up work from the same group which was published in Nature Electronics (doi.org/10.1038/s41928-019-0271-9) last year. Here, they use Si_{0.97}Ge_{0.03} to enhance the material's TE performance and also optimize the μ TEG's power performance by device physics and circuit engineering principles. Besides, they show the demonstrations of energizing integrated circuits by the μ TEGs. The article is well organized, and I only have a few suggestions for improvements.

1. I suggest the power-generating performance of μ TEGs under higher ΔT should be shown, especially for the harvest mode one, because higher ΔT s are used in the demonstrations.
2. For me the explanation of the thermal circuit is confusing. Especially reading the description of fig 2 c, where it is said that the heat flow is in the plane of the paper. It would seem that it is both thermally and electrically in series (maybe there is more information in the supporting information). A visual indication of the heat flow in the images would be helpful.
3. For the powering of low-power electronic devices an output potential of least 50 mV is required. The authors should at least present a concept how to achieve higher output potential with their microdevice.

Reviewer #3 (Remarks to the Author):

Extremely interesting paper on Silicon related thermoelectricity that abounds in the architecture presented previously by the authoring group. The authors report small area TEGs ($< 1\text{mm}^2$) using standard Si IC processing. Low dimensional Si in the shape of blades are used as thermoelectric material. Despite poor thermoelectric performance in terms of usual figure of merit, ZT, the authors show that for an operational point of view comparable power densities can be obtained to those of conventional, hard to miniaturize, modules and that the obtained power densities are good enough to power IoT devices even at ΔT s of 15-25 K. The ability of integrating a very large number of thermocouples with good control of parasitic (thermal and electric) resistances overcome their poor individual performance.

The volume of involved devices and analysed data leads to a nice corpus of experimental evidence that sustain the main conclusions of the work.

As commented earlier, the basic device structure and associated technological workflow has been already introduced in a previous paper. Two are the main novelties of the present manuscript. One is that their nanoblade-shaped active material is now SiGe instead of Si, with a max 3% of Ge content introduced by ion implantation. The other is the extension of device characterization to harvesting scenarios by directly heating the devices with an external source rather than using a built-in heater as in the previous paper.

Some constructive criticisms follow:

Ge content and SiGe material/TEGs:

SiGe alloys are considered by the authors because a lower thermal conductivity is expected for them. Earlier evidence (not referenced) pointed to significant thermal effects when %Ge was in the 20-80% range. A literature value is showcased in the text for a 40% Ge. However, the authors are using a much lower Ge content. The apparent efficacy of such lower content is worth a comment. The included references deal mostly with SiGe bulk or nanostructured bulk values, but no references are given for SiGe material in nanowire form, either experimental or theoretical, nor are recent papers on SiGe NWs TEGs considered for completing the picture.

Ex:

- H.Kim, I.Kim, H. Choi, W. Kim, Thermal conductivities of $\text{Si}_{1-x}\text{Ge}_x$ nanowires with different germanium concentrations and diameters, *Applied Physics Letters* 96 (2010) 233106 <https://doi.org/10.1063/1.3443707Z>
- Z. Wang and N. Mingo, Diameter dependence of SiGe nanowire thermal conductivity, *Applied Physics Letters* 97 (2010) 101903, <https://doi.org/10.1063/1.3486171>
- M. Amato, M. Palummo, R. Rurali and S. Ossicini, Silicon–Germanium Nanowires: Chemistry and Physics in Play, from Basic Principles to Advanced Applications, *Chem. Rev.* 114 (2014) 1371–1412, <https://dx.doi.org/10.1021/cr400261y>
- J.A. Pérez-Taborda, O. Caballero-Calero and M. Martín-González, Silicon-Germanium (SiGe) Nanostructures for Thermoelectric Devices: Recent Advances and New Approaches to High Thermoelectric Efficiency - Chapter 8 in *New Research on Silicon - Structure, Properties, Technology from Intechopen*, 2017, <http://dx.doi.org/10.5772/67730>
- J. Li, Q. Xiang, R. Ze, M. Ma, S. Wang, Q. Xie, Y. Xian, Thermal and electrical analysis of SiGe thermoelectric unicycle filled with thermal insulation materials, *Applied Thermal Engineering* 134 (2018) 266-274, <https://doi.org/10.1016/j.applthermaleng.2018.01.100>
- I. Donmez-Noyan, G. Gadea, M. Salleras, M. Pacios, C. Calaza, A. Stranz, M. Dolcet, A. Morata, A.

Tarancon, L. Fonseca, SiGe nanowire arrays based thermoelectric microgenerator, *Nano Energy* 57 (2019) 492–499, <https://doi.org/10.1016/j.nanoen.2018.12.050>

Have the different Ge contents been measured/analyzed by any physical characterization mean or are they estimations coming from implantation simulations? First order out-of-envelope calculations from the reported Ge implantation doses seem to be insufficient for a homogeneous 3% Ge content in the upper 0.5 micron of the implanted silicon. Post-anneal Ge concentration estimated curves could be a useful addition to the supplementary material to clarify this point.

Test devices and harvest devices:

'Test devices' are said to be designed for power density and 'harvest devices' for voltage density. However, at a first glimpse of Fig 1 and Fig 2, power obtained for the former is lower than the one for the latter. Of course, the difference is the area in each case. One learns later that the test device is indeed a single thermocouple device with a minimum area while the harvest device features hundreds of them (yet in an impressive small area). My point is that this issue should be more clearly stated when introducing Fig 1 and 2, and that power densities and/or voltage densities should be used for the comparison, either graphically or through direct statements in the text, so that the differences in design reveal themselves.

The authors state that sixteen different designs have been produced for the test devices. One parameter was Ge content. 3% Ge was found to yield the best power densities within a given range. Could the authors comment on which was the optimum combination of the other design parameters that led to the maximum obtained value (30uW/cm²/K²)?

In the previous paper, the authors were able to calculate/estimate the 'real' ΔT perceived by the nanoblades. Have they done so in the current experiment and have estimated if the fraction of the ΔT perceived by the nanoblades increased with Ge content?

The resistance of the test mode / power optimum device is around 5 ohms, and the one of the corresponding unit cell for the harvest mode /voltage optimum device is about 120 ohms. Do the authors know which part of it corresponds to the nanoblade arrangement and which to the rest of concurring elements (Cu lines, W plugs, Si wells/silicide)? Do the device resistance change with Ge content?

Powering IoT devices:

Interestingly, the authors have powered in a lab setup some IoT related devices: a PMIC and an optical sensor. Moreover, they have succeeded to do it with moderate values for an external ΔT . How severe is, anyway, the incapacity of powering through the BQ25570 PMIC devices with $R_L < 0.9\text{Mohms}$? What would that suppose in a realistic scenario: total impossibility of powering, need of a secondary battery, duty cycle compromises...?

It may be useful to ponder why the OPT3001 needs a 22K ΔT (2.2 V) when powered directly by the TEG and it would not work with a ΔT of 17.5K, which produces 1.8V, if that is the voltage supplied by a TEG powered BQ25570?

Others:

The devices are made with a commercial 65nm CMOS technology. 80 nm was chosen though as the minimum feature for the nanoblades. Is then 80nm the lowest reliable possible dimension in that technology?

In the introduction, biomedical applications (including implants) and embedded IoT sensors are mentioned as devices welcoming autonomous power other than batteries. Although (very) small footprints harvesters would indeed be welcome by such applications, they are specially challenging for TEGs from a thermal scenario point of view and effective ΔT capture. Do the authors believe that those applications are really the lowest hanging fruit for the thermoelectric generators they are after?

Harvesting characterization is the characterization mode that more closely resembles operational conditions. Moving from test mode (built-in heater) into harvesting mode using an external hot object as a heat source is a step forward. However, the cold side operation is still a bit unrealistic since in practice the whole probe station is used as heat exchanger to the ambient at that end, probably pinning down T_c to a much lower value that would be developed by itself. Have the authors tried to heat the device with a hotplate/thermal chuck and attach a regular size heat exchanger to the Al pad?

Typos/errors:

In line 109, it is wrongly said that a given commercial TEG (ref 22) produces 2.71 W for $T_h = 110\text{ }^\circ\text{C}$ and $T_c = 50\text{ }^\circ\text{C}$. The power delivered in those conditions is 0.41 W according to the datasheet, as correctly pointed out in line 169

References 35 and 36 has gone misplaced after the supplementary information.

In the supplementary information previous authors work is wrongly referenced as 22 instead of 23.

Luis Fonseca

In the following, the Reviewers' verbatim comments are quoted in *black italic font*. The authors' response is in *blue font*.

Response to Reviewer #1:

This is a sequel to the authors' previous work of Ref.23 and 25. The novelties of this work are the inclusion of Ge into Si thermoelements and the demonstration operation of their micro-TEG device by an external heat source to drive off-the-shelf power management IC and light sensor. I can support the authors' approach to optimize the power density at given temperature difference, rather than focusing on thermodynamic efficiency. This paper will attract interest of scientists in the areas of thermoelectric energy harvesting.

It is a notable result that the specific power density was improved significantly by doping small amount of Ge up to 3 % in Si blade. However, it is to be regretted that there is no discussion on the scientific reasoning. I think there is no reason not to show the dependence of Seebeck coefficient on the Ge concentration. I expect that the authors provided test element group devices to measure the resistivity and carrier concentration of SiGe blades. These data are essential to deepen understanding of the effect of Ge doping.

Lines 95-102 (“Si_{1-x}Ge_x was used because ... using only a few % Ge.”) of the revised manuscript expand the explanation of the scientific reasoning for why only a few % Ge in Si improves TEG performance significantly. The basic explanation is that the thermal conductivity κ for Si_{1-x}Ge_x decreases with increasing x very strongly in the narrow range from $x = 0$ to $x = 0.05$, where the thermopower and resistivity change very little with x . This causes a strong increase in TE figure-of-merit and hence performance. References 25, 30–33 have been added to support this explanation. See also our response to Reviewer #3's first comment.

The revised Fig. 1d adds the dependence of the measured Seebeck coefficient on the Ge concentration in otherwise identically structured TEG devices.

I am interested in the statement in line 52, “It has proven difficult to microfabricate high ZT materials...” The authors might want to mention the reason for this statement, or show supporting references.

Lines 56-58 (“Small area ... and moderate $\Delta T \sim 20$ K”) of the revised manuscript re-write this statement to clarify our meaning. We now state the fact that high ZT TEGs having areas < few mm² have been unable to reach necessary voltage and current generation thresholds. References 4, 6, and 7 have been cited to substantiate this statement. In particular, Ref. 7 includes a detailed Table showing that TEGs with areas $\ll 1$ cm² have not yet been able to exceed 1 V with μ W of power operating from $\Delta T \approx 20$ K.

Response to Reviewer #2:

The design, fabrication, and analysis of performance keypoints is innovative, under my point of view. All these things are meaningful and very different from the classical TE-devices.

The manuscript is a follow-up work from the same group which was published in Nature Electronics (doi.org/10.1038/s41928-019-0271-9) last year. Here, they use Si_{0.97}Ge_{0.03} to enhance the material's TE performance and also optimize the μ TEG's power performance by device physics and circuit engineering principles. Besides, they show the demonstrations of energizing integrated circuits by the μ TEGs. The article is well organized, and I only have a few suggestions for improvements.

1. *I suggest the power-generating performance of μ TEGs under higher ΔT should be shown, especially for the harvest mode one, because higher ΔT s are used in the demonstrations.*

Fig. 3 has been replaced with new data showing harvest mode μ TEG performance up to $\Delta T = 23.2$ K, which is in the range of ΔT s used in the device demonstrations shown in Fig. 4.

2. *For me the explanation of the thermal circuit is confusing. Especially reading the description of fig 2 c, where it is said that the heat flow is in the plane of the paper. It would seem that it is both thermally and electrically in series (maybe there is more information in the supporting information). A visual indication of the heat flow in the images would be helpful.*

We think Reviewer 2 misread the caption of Fig. 2c in the original manuscript; the caption stated that "Heat flows into (emphasis added) the plane of the page as shown."

To minimize potential for confusion, Fig. 2c in the revised version now adds an "into the page" vector symbol marked as "Heat flow Q into page", and its caption has been re-written to describe the relative directions of heat flow and electrical current.

3. *For the powering of low-power electronic devices an output potential of least 50 mV is required. The authors should at least present a concept how to achieve higher output potential with their microdevice.*

We do not understand why Reviewer 2 wrote this comment. In the section around Fig. 4 in the original manuscript (retained in the revised manuscript), we clearly show that our harvest mode μ TEGs can deliver an output potential of ≥ 1.8 V with sufficient current to successfully power two examples of commercial low-power electronic devices. This is a central result of this paper.

Response to Reviewer #3:

Extremely interesting paper on Silicon related thermoelectricity that abounds in the architecture presented previously by the authoring group. The authors report small area TEGs ($< 1\text{mm}^2$) using standard Si IC processing. Low dimensional Si in the shape of blades are used as thermoelectric material. Despite poor thermoelectric performance in terms of usual figure of merit, ZT, the authors show that for an operational point of view comparable power densities can be obtained to those of conventional, hard to miniaturize, modules and that the obtained power densities are good enough to power IoT devices even at ΔT s of 15-25 K. The ability of integrating a very large number of thermocouples with good control of parasitic (thermal and electric) resistances overcome their poor individual performance.

The volume of involved devices and analysed data leads to a nice corpus of experimental evidence that sustain the main conclusions of the work.

As commented earlier, the basic device structure and associated technological workflow has been already introduced in a previous paper. Two are the main novelties of the present manuscript. One is that their nanoblade-shaped active material is now SiGe instead of Si, with a max 3% of Ge content introduced by ion implantation. The other is the extension of device characterization to harvesting scenarios by directly heating the devices with an external source rather than using a built-in heater as in the previous paper.

Some constructive criticisms follow:

Ge content and SiGe material/TEGs:

SiGe alloys are considered by the authors because a lower thermal conductivity is expected for them. Earlier evidence (not referenced) pointed to significant thermal effects when %Ge was in the 20-80% range. A literature value is showcased in the text for a 40% Ge. However, the authors are using a much lower Ge content. The apparent efficacy of such lower content is worth a comment. The included references deal mostly with SiGe bulk or nanostructured bulk values, but no references are given for SiGe material in nanowire form, either experimental or theoretical, nor are recent papers on SiGe NWs TEGs considered for completing the picture.

As in our response to Reviewer #1's similar comment, Lines 95-102 ("Si_{1-x}Ge_x was used because ... using only a few % Ge.") of the revised manuscript expand the explanation of the reasoning for why only a few % Ge in Si improves TEG performance significantly. The basic explanation is that the thermal conductivity κ for Si_{1-x}Ge_x decreases with increasing x very strongly in the narrow range from $x = 0$ to $x = 0.05$, where the thermopower and resistivity change very little with x . This causes a strong increase in TE figure-of-merit and hence performance. References 25, 30–33 have been added to support this explanation. The graph below, cited in the revised manuscript as Ref. 30, makes this point:

[Redacted]

(from M. Wagner, *Simulation of Thermoelectric Devices*. Ph.D. Dissertation, Tech. Univ. Wien, Nov. 2007)

The revised manuscript adds References 25,27-29, and 33 on recent SiGe nanowire TEG work. We thank Reviewer #3 for pointing out to us numerous such references.

Have the different Ge contents been measured/analyzed by any physical characterization mean or are they estimations coming from implantation simulations? First order out-of-envelope calculations from the reported Ge implantation doses seem to be insufficient for a homogeneous 3% Ge content in the upper 0.5 micron of the implanted silicon. Post-anneal Ge concentration estimated curves could be a useful addition to the supplementary material to clarify this point. The Ge contents have been estimated by implantation simulations. We re-did implant simulations using two simulation tools: TRIM and TCAD. Reviewer #3 is correct in estimating that the implant parameters given in the original manuscript are insufficient for a homogeneous 3% Ge content in the upper 0.5 micron of silicon – this was based on an inaccurate previously existing simulation.

Lines 261-269 (“Simulations of Ge density ... to a nominal depth of 350 nm.”) of the revised manuscript add more details about the implantation simulations and results, with the addition of Refs. 42, 43, and 44. A graph of simulated Ge concentrations vs. depth into Si, both as-implanted and post-annealed, has been added as Fig. S1 in the revised Supplementary Materials.

A typographical error in the original manuscript, which listed the Ge ion dose at 270 keV as $1.6 \times 10^{16} \text{ cm}^{-2}$, has been corrected. The correct dose used was $2.4 \times 10^{16} \text{ cm}^{-2}$.

Test devices and harvest devices:

‘Test devices’ are said to be designed for power density and ‘harvest devices’ for voltage density. However, at a first glimpse of Fig 1 and Fig 2, power obtained for the former is lower than the one for the latter. Of course, the difference is the area in each case. One learns later that the test device is indeed a single thermocouple device with a minimum area while the harvest device features hundreds of them (yet in an impressive small area). My point is that this issue should be more clearly stated when introducing Fig 1 and 2, and that power densities and/or voltage densities should be used for the comparison, either graphically or through direct statements in the text, so that the differences in design reveal themselves.

Lines 75-87 (“Each test mode ... voltage density rather than power.”) of the revised manuscript have been re-written to explicitly and clearly describe the different functions of “test” and “harvest” mode devices right at the beginning of the Results section. In particular, we now state up front that the test mode devices were designed to optimize power density, not voltage, while the harvest mode devices were designed to maximize voltage density, not power.

The authors state that sixteen different designs have been produced for the test devices. One parameter was Ge content. 3% Ge was found to yield the best power densities within a given range. Could the authors comment on which was the optimum combination of the other design parameters that led to the maximum obtained value (30uW/cm2/K2)?

We think Reviewer #3 mis-interpreted our statement that 16 different designs were produced for the test devices. From what Reviewer #3 wrote, it appears that he/she thought that the 16 different designs included variations of Ge content x . This is not correct. For each $x = 0, 0.01, 0.02, \text{ and } 0.03$, we tested 16 different device layouts having the same x but different structural and geometric parameters. Thus if we included x as a parameter, we tested $16 \times 4 = 64$ different test mode μ TEG variants.

Lines 128-135 (“For each value of x ... gave Γ_P between 20 to 30 $\mu\text{Wcm}^{-2}\text{K}^{-2}$.”) of the revised manuscript have been re-written to make our explanation of this matter clearer and hopefully less open to mis-interpretation by a reader.

Lines 135-140 (“Higher Γ_P layouts were associated ... to decrease the thermopile’s R_S .”) of the revised manuscript add some comments about what combination of design parameters maximized power generation performance.

In the previous paper, the authors were able to calculate/estimate the ‘real’ ΔT perceived by the nanoblades. Have they done so in the current experiment and have estimated if the fraction of the ΔT perceived by the nanoblades increased with Ge content?

Lines 124-131 of the revised manuscript have been added to answer this question.

The resistance of the test mode / power optimum device is around 5 ohms, and the one of the corresponding unit cell for the harvest mode /voltage optimum device is about 120 ohms. Do the authors know which part of it corresponds to the nanoblade arrangement and which to the rest of concurring elements (Cu lines, W plugs, Si wells/silicide)? Do the device resistance change with Ge content?

Lines 176-180 (“If we scale ... in the test mode device.”) of the revised manuscript have been added to answer this question. After scaling for the different number of TE blade elements in parallel, the harvester’s scaled resistance per thermocouple is slightly higher than the test mode’s scaled resistance per thermocouple. We cannot be certain where the extra resistance comes from, but we do know that the harvest mode devices have extra metallization per thermocouple in order to connect the thermocouples in series.

Line 119 of the revised manuscript adds a statement that the source resistance of a test mode device increases by $\sim 10\%$ going from $x = 0$ to $x = 0.03$.

Powering IoT devices:

Interestingly, the authors have powered in a lab setup some IoT related devices: a PMIC and an optical sensor. Moreover, they have succeeded to do it with moderate values for an external ΔT . How severe is, anyway, the incapacity of powering through the BQ25570 PMIC devices with $RL < 0.9\text{Mohms}$? What would that suppose in a realistic scenario: total impossibility of powering, need of a secondary battery, duty cycle compromises...?

Lines 195-199 (“If this $\mu\text{TEG}/\text{PMIC}$ configuration ... backup battery with the PMIC.”) of the revised manuscript have been added to answer this question.

It may be useful to ponder why the OPT3001 needs a 22K ΔT (2.2 V) when powered directly by the TEG and it would not work with a ΔT of 17.5K, which produces 1.8V, if that is the voltage supplied by a TEG powered BQ25570?

Lines 205-209 (“This was the smallest ... but at zero current.”) of the revised manuscript have been added to answer this question. In short, a ΔT near 17.5 K would generate an open-circuit voltage of 1.8 V, but at zero current. The smallest ΔT needed to generate both sufficient voltage and current was near 22 K.

Others:

The devices are made with a commercial 65nm CMOS technology. 80 nm was chosen though as the minimum feature for the nanoblades. Is then 80nm the lowest reliable possible dimension in that technology?

Lines 282-283 (“An 80 nm width was used ... “65 nm node” process technology.”) of the revised manuscript have been added to answer this question.

In the introduction, biomedical applications (including implants) and embedded IoT sensors are mentioned as devices welcoming autonomous power other than batteries. Although (very) small footprints harvesters would indeed be welcome by such applications, they are specially challenging for TEGs from a thermal scenario point of view and effective ΔT capture. Do the authors believe that those applications are really the lowest hanging fruit for the thermoelectric generators they are after?

Reviewer #3 is correct to question whether biomedical applications are “...the lowest hanging fruit...” for our μ TEGs. In fact, biomedical applications constitute perhaps the most challenging application for TEGs because they generally involve small ΔT s and difficult thermal interface conditions.

The revised manuscript removes references to possible biomedical applications in both the Abstract and the opening paragraph. Instead, Reference 2 has been added as it describes a range of “low hanging fruit” potential applications for TEGs.

Lines 234-247 (“Assuming thermal interface issues ... of this harvesting mode μ TEG design.”) of the revised manuscript add a paragraph to comment on μ TEG performance parameters needed for biomedical applications, and what it might take to achieve such performance.

Harvesting characterization is the characterization mode that more closely resembles operational conditions. Moving from test mode (built-in heater) into harvesting mode using an external hot object as a heat source is a step forward. However, the cold side operation is still a bit unrealistic since in practice the whole probe station is used as heat exchanger to the ambient at that end, probably pinning down T_c to a much lower value that would be developed by itself. Have the authors tried to heat the device with a hotplate/thermal chuck and attach a regular size heat exchanger to the Al pad?

Lines 224-233 (“All our μ TEG devices were designed ... directing external heat into an IC chip.”) of the revised manuscript add a paragraph to comment on the importance of thermal interfacing and heat exchange. We state that the devices used for this research were designed to be tested in a wafer probe station and hence were not compatible with standard heat exchangers. While we acknowledge the importance of thermal interfaces in order to use a TEG in any practical application, research on efficient heat exchangers and thermal packaging is another step beyond the scope of this paper, which focuses on device performance.

We have tried Reviewer #3’s suggestion to try reversing the normal hot/cold contacts to a harvest mode μ TEG, *i.e.*, to use a thermal chuck to heat the backside of a μ TEG chip and use the frontside Al thermal pad as the cold reservoir. Because these devices were made to be wafer probed, the frontside Al thermal pad could not be attached to a regular size heat exchanger – we could only

use the small-diameter Cu rod, unheated, touching the Al pad to try to thermally anchor it. However, because the silicon chip area is smaller than the thermal chuck area, we found that heating the chuck resulted in a layer of heated air directly over the silicon chip. This temperature of this air layer was measured to be within 2 to 3 K of the chuck's temperature, so a maximum ΔT of < 3 K could be maintained this way. We do note that the $|V_{OC}|$ then measured from the harvester μ TEG operated in this "reverse" manner was consistent with the $|V_{OC}|$ values obtained the "normal" way, *i.e.*, using the Cu rod as heater and the chuck as room-temperature cold reservoir.

Typos/errors:

In line 109, it is wrongly said that a given commercial TEG (ref 22) produces 2.71 W for $T_h = 110$ °C and $T_c = 50$ °C. The power delivered in those conditions is 0.41 W according to the datasheet, as correctly pointed out in line 169

References 35 and 36 has gone misplaced after the supplementary information.

In the supplementary information previous authors work is wrongly referenced as 22 instead of 23.

These errors have been corrected.

REVIEWERS' COMMENTS:

Reviewer #1 (Remarks to the Author):

The revised manuscript has addressed all the comments. The added Figure 1d, showing the dependence of the Seebeck coefficient on the Ge concentration, is very important to explain the drastic improvement of the specific power density, together with the decrease in the thermal conductivity. The revised manuscript, in my opinion, is suitable for publication.

Reviewer #2 (Remarks to the Author):

I am fully satisfied with the revisions and I suggest to accept the manuscript as it is.

Reviewer #3 (Remarks to the Author):

In the opinion of this reviewer, authors have conveniently addressed the open issues and offered recommendations by directly commenting them in the rebuttal letter and improving the paper text accordingly. Still small remaining issues are no obstacle to publish the manuscript in its current form.

Ge content influence:

OK. Authors' discussion is improved by arguing and supporting the fact that a good deal of SiGe thermal conductivity reduction is already achieved at low Ge contents - similar to the ones employed in the work.

Ge content implanted:

OK. The situation on this issue has been straightened, the text has been corrected and the supplementary information extended accordingly. I wonder if the authors have Z in lines 95-102 on purpose instead of ZT as in the rest of the paper. It is technically correct, but strange.

Test vs harvesting mode:

I think the rewriting has helped to clarify the distinction with respect the *intention* behind the two designs. Still, I think that you could do better with respect the *results* by, for instance, giving the reader the cue that the test mode design yields 6 times more power per unit area, and the harvest mode device yields 3.6 times more voltage per unit area. Otherwise, direct comparison of Fig 3 and Fig 1 just show that the harvest mode device offers more voltage and power than the test mode device. In any case, I do not deem necessary to modify further the paper, but if by chance you need to make a new revision, you may consider this point.

16 different designs:

OK. According to authors' answer, I think I probably did not explain myself properly. I was not arguing about the number of different designs, but I was curious about which was the combination of structural and geometric parameters that offered the best test mode performance in addition to the Ge content parameter. What you have written in lines 135-140 gives some insight in that direction, though.

ΔT perceived by nanoblades:

OK. New lines 120-127 help to clarify this point.

Nanoblades resistance:

I was not so intent on comparing test mode and harvesting mode devices but on knowing which part of the internal resistance of the generator corresponds to the low dimensional TE silicon material. Maybe, for the zero Ge content sample, it could be inferred from the doping and the geometry of the nanoblades arrangement. Although interesting, it is not a fundamental aspect of the paper. Knowing that Ge inclusion induces a slight increase of resistance is a nice piece of information. Clearly, the positive thermal aspects of such inclusion outweigh the electric resistance worsening. It agrees with my experience, too.

Powering devices:

OK. Doubts have been handled nicely and the text reinforced.

Others:

OK. Also taken care of appropriately.

Luis Fonseca

Detailed Response to Reviewers for Manuscript NCOMMS-20-15774B:

In the following, the Reviewers' verbatim comments are quoted in *black italic font*. The authors' response is in **blue font**.

Response to Reviewer #1:

The revised manuscript has addressed all the comments. The added Figure 1d, showing the dependence of the Seebeck coefficient on the Ge concentration, is very important to explain the drastic improvement of the specific power density, together with the decrease in the thermal conductivity. The revised manuscript, in my opinion, is suitable for publication.

Reviewer #1 recommends publication without further revisions. No response is necessary.

Response to Reviewer #2:

I am fully satisfied with the revisions and I suggest to accept the manuscript as it is.

Reviewer #2 recommends publication without further revisions. No response is necessary.

Response to Reviewer #3:

In the opinion of this reviewer, authors have conveniently addressed the open issues and offered recommendations by directly commenting them in the rebuttal letter and improving the paper text accordingly. Still small remaining issues are no obstacle to publish the manuscript in its current form.

Ge content influence:

OK. Authors' discussion is improved by arguing and supporting the fact that a good deal of SiGe thermal conductivity reduction is already achieved at low Ge contents - similar to the ones employed in the work.

No response is necessary.

Ge content implanted:

OK. The situation on this issue has been straightened, the text has been corrected and the supplementary information extended accordingly. I wonder if the authors have Z in lines 95-102 on purpose instead of ZT as in the rest of the paper. It is technically correct, but strange.

No response is necessary.

Test vs harvesting mode:

*I think the rewriting has helped to clarify the distinction with respect the **intention** behind the two designs. Still, I think that you could do better with respect the **results** by, for instance, giving the reader the cue that the test mode design yields 6 times more power per unit area, and the harvest mode device yields 3.6 times more voltage per unit area. Otherwise, direct comparison of Fig 3 and Fig 1 just show that the harvest mode device offers more voltage and power than the test mode device. In any case, I do not deem necessary to modify further the paper, but if by chance you need to make a new revision, you may consider this point.*

We like Reviewer #3's suggestion to cue the reader that the test and harvest mode devices do deliver the intended different power and voltage density **results. Therefore, near the bottom of p. 4 of the final manuscript we added: "As the following results show, operating from nearly the**

same T_C and T_H , test mode devices generated power density $\sim 6x$ higher than harvest mode, while harvest mode devices generated voltage density $\sim 3.6x$ higher than test mode.”

16 different designs:

OK. According to authors' answer, I think I probably did not explain myself properly. I was not arguing about the number of different designs, but I was curious about which was the combination of structural and geometric parameters that offered the best test mode performance in addition to the Ge content parameter. What you have written in lines 135-140 gives some insight in that direction, though.

No response is necessary.

ΔT perceived by nanoblades:

OK. New lines 120-127 help to clarify this point.

No response is necessary.

Nanoblades resistance:

I was not so intent on comparing test mode and harvesting mode devices but on knowing which part of the internal resistance of the generator corresponds to the low dimensional TE silicon material. Maybe, for the zero Ge content sample, it could be inferred from the doping and the geometry of the nanoblades arrangement. Although interesting, it is not a fundamental aspect of the paper. Knowing that Ge inclusion induces a slight increase of resistance is a nice piece of information. Clearly, the positive thermal aspects of such inclusion outweigh the electric resistance worsening. It agrees with my experience, too.

As Reviewer #3 states, this issue is not a fundamental aspect of this paper. In fact, which part of the internal resistance corresponds to the TE silicon material and which part to parasitic resistances was addressed by us in two previous publications (Refs. 22 and 24) for the zero Ge content ($x = 0$) devices. As readers of this paper may be curious about this issue, near the middle of p. 5 we added the following sentence to point readers to these previous results: “Previous modeling^{22,24} of $x = 0$ test mode devices estimated the parasitic resistance from leads and contacts to be $\sim 2 \Omega$ per thermopile.”

Powering devices:

OK. Doubts have been handled nicely and the text reinforced.

No response is necessary.

Others:

OK. Also taken care of appropriately.

No response is necessary.